# Enhanced Antibacterial Activity of Substituted Derivatives of NCR169C Peptide

**DOI:** 10.3390/ijms24032694

**Published:** 2023-01-31

**Authors:** Dian H. O. Howan, Sándor Jenei, János Szolomajer, Gabriella Endre, Éva Kondorosi, Gábor K. Tóth

**Affiliations:** 1Department of Medical Chemistry, Albert Szent-Györgyi Medical School, University of Szeged, H-6720 Szeged, Hungary; 2Biological Research Centre, Institute of Plant Biology, H-6726 Szeged, Hungary; 3MTA-SZTE Biomimetic Systems Research Group, Albert Szent-Györgyi Medical School, University of Szeged, H-6720 Szeged, Hungary

**Keywords:** antimicrobial resistance, antimicrobial peptides, *Medicago truncatula*, nodule-specific cysteine-rich, modified trytophans

## Abstract

*Medicago truncatula* in symbiosis with its rhizobial bacterium partner produces more than 700 nodule-specific cysteine-rich (NCR) peptides with diverse physicochemical properties. Most of the cationic NCR peptides have antimicrobial activity and the potential to tackle antimicrobial resistance with their novel modes of action. This work focuses on the antibacterial activity of the NCR169 peptide derivatives as we previously demonstrated that the C-terminal sequence of NCR169 (NCR169C_17–38_) has antifungal activity, affecting the viability, morphology, and biofilm formation of various *Candida* species. Here, we show that NCR169C_17–38_ and its various substituted derivatives are also able to kill ESKAPE pathogens such as *Enterococcus faecalis*, *Staphylococcus aureus*, *Klebsiella pneumoniae*, *Acinetobacter baumannii*, *Pseudomonas aeruginosa*, and *Escherichia coli.* The replacement of the two cysteines with serines enhanced the antimicrobial activity against most of the tested bacteria, indicating that the formation of a disulfide bridge is not required. As tryptophan can play role in the interaction with bacterial membranes and thus in antibacterial activity, we replaced the tryptophans in the NCR169C_17–38_C_12,17_/S sequence with various modified tryptophans, namely 5-methyl tryptophan, 5-fluoro tryptophan, 6-fluoro tryptophan, 7-aza tryptophan, and 5-methoxy tryptophan, in the synthesis of NCR169C_17–38_C_12,17_/S analogs. The results demonstrate that the presence of modified fluorotryptophans can significantly enhance the antimicrobial activity without notable hemolytic effect, and this finding could be beneficial for the further development of new AMPs from the members of the NCR peptide family.

## 1. Introduction

Antimicrobial resistance (AMR) is a condition in which pathogenic microorganismsdevelop a survival ability against exposure to medicine that would normally demolish them or inhibit their growth [1]. This condition has been observed since the first finding of antibiotics and AMR has developed rapidly due to the excessive use of antibiotics in healthcarelsystems in recent decades after the ‘golden era’ of antibiotics [2]. The World Health Organization (WHO) declared AMR as among the top 10 international public health threats and, together with authorities all around the globe, initiated a program called the GLASS (global antimicrobial resistance and use surveillance system) in 2015 as a response to this phenomenon [3]. The WHO has also made a list of global priority pathogens, in which the so-called ESKAPE group, consisting of *Enterococcus faecalis*, *Staphylococcus aureus*, *Klebsiella pneumoniae*, *Acinetobacter baumannii*, *Pseudomonas aeruginosa*, and *Escherichia coli*, is on the top of the list as priority bacteria that can “escape” conventional antibiotics and medication, causing multi-drug resistance (MDR) [4,5].

More than 2 million people in the US are affected by AMR infections and the mortality rate caused by it was approximately 700,000 in 2014 [1,6]. This number is predicted to gradually increase by 14-fold in 2050 [1]. During the Coronavirus 2019 (COVID-19) pandemic, heightened potential threats caused by AMR emerged due to the high dependency of COVID-19 management on pharmacological interventions, which hastened the evolution and the expansion of AMR [7,8]. For example, many patients with mild diseases with or without pneumonia receive antibiotics [9]. Another report has revealed that among 72% of patients who received antibiotics treatment, 8% showcased superimposed bacterial or fungal co-infections [10]. This current condition needs serious intervention, including the discovery of new antimicrobial agents that could combat this resistance issue [11].

Antimicrobial peptides (AMPs) are among the candidates with the most potential for the treatment of infectious diseases and immunotherapies with a low chance of developing resistance against bacterial strains [12,13]. AMPs have preserved their ability to kill microbes, including bacteria, archaea, fungi, yeasts, and viruses, for millions of years because this group of peptides can exploit fundamental features of the microbial cellular membrane [14,15,16,17,18]. AMPs or host-defense peptides (HDPs) have long been a frontier in the innate immune system of living organisms [14,19,20], with their unique characteristics of short amino acid sequences, mostly positive net charges, average hydrophobic content of approximately 43%, and broad spectra of antimicrobial activity [14,15,16,17]. In addition to their antimicrobial activity, AMPs have also displayed inhibition activity against the formation of microbial biofilms, which are related to the majority of all infectious diseases in humans [12]. The most recent number of AMPs from the antimicrobial peptide database (APD) in 2020 was 3257 peptides, of which seven peptides have been approved by the U.S. Food and Drug Administration (FDA) [13,21]. Some AMPs, such as Bacitracin, Polymyxins, Daptomycin, Vancomycin, and Gramicidin, have already been used clinically [22]. The small number of AMP-based drugs on the market nowadays might be due to the low selectivity of AMPs against bacteria and their toxicity against eukaryotic cells, as well as the short half-life of these peptides in vivo [12,22].

The mechanisms of action of AMPs are diverse. Mostly, they are bactericides and interact with bacterial membranes. They can form transient pores or disrupt the membrane, leading to cell lysis [19]. The peptides transferred inside the bacteria can have various intracellular targets, such as DNA, RNA, and proteins [14]. Based on the principal bacterial target, AMPs are grouped into three broad classes: (i) AMPs with intracellular targets, (ii) AMPs with the bacterial membrane as their targets, and (iii) AMPs that activate their mechanism of action through a receptor- and signaling-based response, which allows the bacteria to resist and survive AMP challenge. The resistance mechanisms against groups (ii) and (iii) are varied, from neutralization of the negative bacterial surface charge to deactivation of AMPs by proteolytic degradation [20]. 

AMPs can be classified into two categories based on the abundance of amino acids in their sequences, structures, and biological functions of these peptides. One of these groups is linear molecules, which consist of an α-helical structure without cysteine, e.g., cecropin and magainin, or are rich in certain amino acids, such as proline, glycine, arginine, histidine, and tryptophan [16,22,23]. Another group includes cysteine-containing peptides, which can be both cationic and anionic, and are stabilized by disulfide bonds, which may be important for antimicrobial activity, such as defensins, which represent the largest group of antimicrobial peptides in plants [24]. 

Nodule-specific cysteine-rich (NCR) peptides have evolved in certain clades of leguminous plants that play exclusive roles in symbiosis with *Rhizobium* bacteria and mediate bacterial differentiation into nitrogen-fixing bacteroids in root nodules [25]. NCRs are structurally related to defensins but represent a unique family of peptides with two subgroups containing either four or six cysteine residues, and cysteines are essential for their symbiotic functions [26]. The mature NCR peptides usually have 30–50 amino acids in their sequence, with four or six cysteine residues at the conserved positions in otherwise very diverse amino acid sequences, providing physicochemical properties. These NCRs are only expressed in the symbiotic cells, but in different sets at different stages of the nitrogen-fixing nodule development [27]. In the model legume, *Medicago truncatula*, more than 700 NCR peptides are produced in symbiosis, and numerous exhibit in vitro antimicrobial activity and no or negligible toxicity against human cells [27,28,29]. For example, the 24-amino-acid-long NCR247 peptide, together with its derivates, displayed promising antimicrobial activity against several bacterial strains including ESKAPE pathogens [28]. A recent publication from our group revealed that among 104 synthetic NCR peptides from *M. truncatula*, *M. sativa*, *Pisum sativum*, *Galega orientalis*, and *Cicer arietinum*, almost all cationic peptides displayed a broad antimicrobial spectrum, as well as the effective and rapid killing of ESKAPE bacteria within a minimal bactericidal concentration (MBC) range of 0.8–3.1 µM [30].

NCR169, a cationic peptide (net charge at pH7 is 1.91) consisting of 38 amino acids with four cysteine residues, is crucial for bacteroid differentiation [26], and its absence results in no nitrogen fixation but premature nodule senescence [20,29,31,32,33]. Two NMR structures of NCR169 produced in *Escherichia coli* resulting from different disulfide bond patterns have been determined by utilizing the ^1^H-^15^N heteronuclear single-quantum coherence (HSQC) NMR technique [32]. Both structures had a consensus C-terminal short antiparallel β-sheet, while the extended N-terminal region had different properties; one was mobile, and the other was not. The disulfide bonds of NCR169 contributed to its structural stability and solubility. One of the NCR169 oxidized forms bound to negatively charged bacterial phospholipids. Furthermore, the positively charged lysine-rich region of NCR169 showed membrane binding ability and antimicrobial activity [32]. In a recent publication by Szerencsés et al. [27], the C-terminal sequence of NCR169, NCR169C_17–38,_ and its derivative, in which the two tryptophan residues (W_10_ and W_20_) were replaced with alanine (NCR169C_17–38_W_10,20_/A), were also shown to possess antifungal activity against different *Candida* species. The active peptides inhibited both yeast-hypha transformation and biofilm formation in *Candida*.

In this research, we investigated and demonstrated the antibacterial activity of NCR169C_17–38,_ against various Gram-positive and Gram-negative ESKAPE pathogenic bacteria. Furthermore, we explored the roles of cysteines and tryptophans, as well as different modifications of tryptophans in this sequence, which proved to be important in the antimicrobial activity of the peptides.

## 2. Results

### 2.1. NCR169C and Its Derivates Are Potential Antimicrobial Agents

We synthesized NCR169C_17–38_ with C-terminal amidation and its four derivatives with different substitutions listed in Table 1A. These substitutions have little effect on the physicochemical properties of the peptides (Table 1B). The exchange of the two tryptophans to alanine led to a moderate decrease in hydrophobicity (less than 10%) and changing the two cysteines to serine or alanine led to only a 0.5 increase in the calculated pI value, the GRAVY and the Boman index in all cases have modest values. Furthermore, the 3D structures obtained with AlphaFold predictions of NCR169C_17–38_ and all its derivatives resulted in a similar alpha-helical structure instead of the β-sheet structure present in the C-terminal part of the full-length peptide (Appendix A). All peptides possessed antimicrobial activities against ESKAPE bacteria—*Enterococcus faecalis*; *Staphylococcus aureus*; *Klebsiella pneumoniae*; *Acinetobacter baumannii*; *Pseudomonas aeruginosa*; *Escherichia coli*—as well as *Listeria monocytogenes* and *Salmonella enterica* (Table 1C). NCR169C_17–38_ was able to kill most of the tested bacteria at 3.1 µM, except *E. faecalis* and *E. coli*, requiring 6.3 and 1.6 µM, respectively. The replacement of the two cysteines with serine residues (NCR169C_17–38_C_12,17_/S) did not impair and even improved the activity in the case of *E. faecalis*, *S. aureus*, *A. baumannii*, and *S. enterica*, indicating that cysteines and the formation of a disulfide bridge are not needed for the antimicrobial activity (Table 1C). Tryptophan is proposed to play a critical role in the activity and interaction of AMPs with bacterial membranes. Exchange of the two tryptophan residues with alanine in NCR169C_17–38_ (NCR169C_17–38_W_10,20_/A), indeed abolished or strongly reduced the activity against *E. faecalis*, *S. aureus*, *S. enterica*, *L. monocytogenes*, and *K. pneumoniae*, whereas it remained effective against *A. baumannii*, *P. aeruginosa* and *E. coli.* The combination of these two types of substitutions (NCR169C_17–38_ W_10,20_/A,C_12,17_/S) or the replacement of both tryptophan and cysteine residues with alanine (NCR169C_17–38_W_10,20_C_12,17_/A) retained or further reduced the activities: both peptides were similarly effective against *P. aeruginosa* and *L. monocytogenes* at 3.1 µM, *S. aureus*, *E. coli*, *S. enterica* at 6.3 µM, and *A. baumannii* (12.5 µM) but NCR169C_17–38_W_10,20_C_12,17_/A became inactive against *K. pneumoniae*, which could be killed by NCR169C_17–38_W_10,20_/A,C_12,17_/S (25 µM). 

To find out whether different strains of bacterial species were similarly or differently sensitive to peptides, we determined the minimum bactericidal concentrations of the two most active peptides, NCR169C_17–38_ and NCR169C_17–38_C_12,17_/S, against two additional strains of *E. coli* (ATCC 25922 and ATCC 35218) and against *S. aureus* (ATCC 25923), which are often used for antibiotic susceptibility assays. All these strains were similarly sensitive to these peptides (Table 1D).

### 2.2. Effect of the Tryptophan Residue Modifications on Antimicrobial Activity 

After finding that replacing the two cysteines with the isosteric serine residues (NCR169C_17–38_C_12,17_/S) increased the antimicrobial effect of the original peptide, we synthesized analogs of NCR169C_17–38_C_12,17_/S peptide to investigate their antimicrobial activity by incorporating some modified tryptophan into the sequence at residue numbers 10 and 20 (Table 2.). The following chemically modified tryptophans were used to test whether they have a significant effect on the antimicrobial properties of the peptide: 5-methyl tryptophan (W^5-Me^), 5-fluoro tryptophan (W^5-F^), 6-fluoro tryptophan (W^6-F^), 7-aza tryptophan (W^7-Aza^), and 5-methoxy tryptophan (W^5-MeO^). 5-fluoro tryptophan was used in the l- and d- enantiomeric forms for the synthesis; therefore, these two peptides are known as NCR169C_17–38_C_12,17_/S-10W^5-F-L^ and NCR169C_17–38_C_12,17_/S-10W^5-F-D^. All other analogs of NCR169C_17–38_C_12,17_/S were synthesized using modified tryptophans in racemic forms followed by the separation of the two forms of peptides, except for NCR169C_17–38_C_12,17_/S-20W^5-MeO^. The separated peptides were labeled with codes I and II, referring to either d- or l-configurations of the substituted tryptophan, which could later be determined if an exciting antimicrobial effect justifies it.

The minimum bactericidal concentration of peptides listed in Table 2 was determined identically as in Table 1C, D against the same bacterial strains (Table 3). 

NCR169C_17–38_C_12,17_/S-10W^5-Me^ I and II have similar antibacterial profiles against most of the pathogenic bacteria tested except for *A. baumannii*, in which NCR169C_17–38_C_12,17_/S-10W^5-Me^ II has slightly better activity (3.1 µM) than NCR169C_17–38_C_12,17_/S-10W^5-Me^ I (6.3 µM). The comparison of the activity of these two peptides with the activity of NCR169C_17–38_C_12,17_/S showed that the original compound had better antibacterial activity on *S. aureus*, *K. pneumoniae*, *E. coli*, and *S. enterica*. The same alteration in tryptophan at position 20 (NCR169C_17–38_C_12,17_/S-10W^5-Me^) had a more drastic effect by significantly reducing its ability to kill most tested pathogens, especially in the case of form I, which had an MBC of 12.5 µM for *E. faecalis* and *K. pneumonia* and 6.3 µM for the rest of bacteria, while form II showed low activity only against *E. faecalis* (25 µM). 

Substitution of tryptophan at position 10 with 5-fluoro-L-tryptophan improved the antibacterial activity against several bacteria. NCR169C_17–38_C_12,17_/S-10W^5-F-L^ was able to kill all tested bacteria with 1.6 µM, with the only exception of *K. pneumoniae* (MBC: 3.1 µM). The same modification with the other racemic form of 5-fluoro-D-tryptophan (NCR169C_17–38_C_12,17_/S-10W^5-F-D^) showed an overall antibacterial activity more similar to the parental peptide, the same as when the tryptophan at position 20 was modified to 5-fluoro-tryptophan. The best antibacterial effects were detected for those peptide analogs in which 6-fluoro-tryptophan was used at position 10. In this case, both synthesized peptides, NCR169C_17–38_C_12,17_/S-10W^6-F^ I and II, exhibited a strong killing ability at a concentration as low as 0.8 µM against several pathogens including *E. faecalis*, *S. aureus*, *A. baumannii*, *P. aeruginosa*, and *E. coli*, which was a unique property among the tested peptide analogs. These two peptides were equally effective against additional strains tested such as *E. coli* ATCC 25922 and ATCC 35218, and *S. aureus* ATCC 25923 (Table 3B). 

The introduction of 7-Aza tryptophan into position 10 of the NCR169C_17–38_C_12,17_/S peptide resulted again in two peptides, NCR169C_17–38_C_12,17_/S-10W^7-Aza^ I and II, which had similar antibacterial activity to each other (mostly at 3.1 µM), and slightly lower than that of the parental peptide. The worst series of MBC values against all tested pathogens was detected when 7-Aza-tryptophan was at position 20 of the parental peptide. In this case, NCR169C_17–38_C_12,17_/S-20W^7-Aza^ I peptide had an MBC of 6.3 µM against most bacteria except *E. faecalis* and *L. monocytogenes* that was 25 µM, while NCR169C_17–38_C_12,17_/S-20W^7-Aza^ II analog exhibited the same MBC value of 12.5 µM against all bacterial strains. 

Peptide analogs were also synthesized by introducing 5-methoxy tryptophan into either position 10 (NCR169C_17–38_C_12,17_/S-10W^5-MeO^ I and II) or position 20 (NCR169C_17–38_C_12,17_/S-20W^5-MeO^ containing both racemic forms) and tested for their antibacterial activity. Interestingly, NCR169C_17–38_C_12,17_/S-10W^5-MeO^ II and NCR169C_17–38_C_12,17_/S-20W^5-MeO^ showed the same overall activity characterized by the MBC of 3.1 µM, while NCR169C_17–38_C_12,17_/S-10W^5-MeO^ I had a slightly stronger ability to kill these bacteria at 1.6 µM. However, they displayed very diverse activity against *P. aeruginosa*: NCR169C_17–38_C_12,17_/S-10W^5-MeO^ II could only kill these bacteria at a concentration of 25 µM while NCR169C_17–38_C_12,17_/S-10W^5-MeO^ I was effective against them at a concentration of 1.6 µM and NCR169C_17–38_C_12,17_/S-20W^5-MeO^ even at a concentration of 0.8 µM.

### 2.3. NCR169C_17–38_ and Its Derivatives Do Not Provoke Hemolysis of Human Red Blood Cells

An oxidized form of NCR169 has been shown to bind to negatively charged bacterial phospholipids [32], so it is possible that the antimicrobial activity of NCR169C and their derivatives may cause membrane damage. However, the interaction of these peptides with bacterial membranes does not necessarily mean membrane damage of human cells. Therefore, we tested the potential hemolytic activity of NCR169C_17–38,_ NCR169C_17–38_C_12,17_/S, and the two most active derivatives, NCR169C_17–38_C_12,17_/S-10W^6-F^I and NCR169C_17–38_C_12,17_/S-10W^6-F^II, against human red blood cells. None of the peptides provoked hemolysis in the range of MBCs and not even up to a concentration of 100 µM except NCR169C_17–38_C_12,17_/S-10W^6-F^II, which caused mild hemolysis at higher concentrations (Figure 1). 

## 3. Discussion

The major aim of this study was to explore the antibacterial activity of the C-terminal region of NCR169, which proved to be effective against various *Candida* species [27]. Here, we show that NCR169C_17–38_ was able to kill all the eight pathogenic bacterial species tested, demonstrating that the C-thermal amino acid sequence is necessary and sufficient for the bactericidal activity. The replacement of cysteine or tryptophan residues with other amino acids, or the use of different chemically modified tryptophan, resulted in certain changes in the antimicrobial activity of the given peptide.

When the cysteine residues were substituted with serine in NCR169C_17–38_, the activity of the peptide was not only preserved, but even improved, indicating that the presence of cysteine, and therefore the formation of disulfide bridge, is not essential for the antimicrobial activity of NCR169C_17–38_. The formation of disulfide bridges was also dispensable in certain antimicrobial peptides and the reduction in disulfide bonds increased the antimicrobial activity [34,35,36,37,38]. The addition of serine in combination with some antibacterial drugs enhanced the antimicrobial activity [39,40,41]. We also investigated the role of tryptophans in the studied peptide, as they may be necessary for the interaction of AMPs with bacterial membranes [42,43,44,45]. In line with this, the substitution of tryptophan residues with alanine in NCR169C_17–38_ impaired the antimicrobial activity in all three derivatives (NCR169C_17–38_W_10,20_/A, NCR169C_17–38_W_10,20_/A,C_12,17_/S, NCR169C_17–38_W_10,20_C_12,17_/A). The MBCs of NCR169C_17–38_ and NCR169C_17–38_C_12,17_/S were in the range of 1.6–3.5 µM against most bacterial species tested. NCR169C_17–38_ and its four derivatives listed in Table 1 share similar physicochemical properties and all have an alpha-helical 3D structure as predicted by AlphaFold (Appendix A) [46]. 

The NCR169C_17–38_ peptide displays a strong ability for alpha-helix formation except for the four N-terminal amino acids and remarkably, the two cysteines are not in proximity, as the formation of the disulfide bridge would destroy the high helicity (Appendix A). The unnecessity of the disulfide bond is also supported by the replacement of cysteines with serines, which resulted in a similar alpha-helical structure (Appendix A) and biological activity. Exchanging the tryptophans for alanine (NCR169C_17–38_W_10,20_/A) or these together with the replacement of the cysteines with serine (NCR169C_17–38_W_10,20_/A,C_12,17_/S), or the substitution of both tryptophans and cysteines to alanine (NCR169C_17–38_W_10,20_C_12,17_/A) did not cause any change in the steric structure (Appendix A). On the other hand, this structure differed from that of the full-length NCR169 peptide, which exhibits largely disordered structures with two short anti-parallel beta-pleated sheets in the C-terminal region predicted by AlphaFold (Appendix A), which otherwise matches well with the published NMR structures of the native conformation [32]. Thus, the alpha-helical structure probably plays a role in the antimicrobial activity of the studied peptides; however, the individual differences in their activities cannot be linked to modifications in the 3D structure, rather the quality of individual amino acids is more important, as shown in the examples above. While the alpha-helical 3D structure was characteristic of NCR169C_17–38_ and its four derivatives, it cannot be generalized to the hundreds of diverse NCR peptides, for example in the case of NCR335, where both the N- and C-terminal part displays antimicrobial activity, one part has helical, while the other has beta-pleated sheet structure [27]. 

Based on these findings, we constructed new NCR169C analogs by incorporating five commercially available modified tryptophans in racemic forms or, in the case of 5-fluoro tryptophan, in L- and D-forms. The introduction of modified amino acids into the peptides generated new chemical and biological properties. 

5-methyl-tryptophan at position 10 resulted in increased MBC (6.3 µM) against two Gram-negative bacteria (*K. pneumoniae* and *A. baumannii*), while for the rest of the tested bacteria, it was effective in 3.1 µM. 5-methyl-tryptophan at position 20 dramatically reduced the activity, thus increasing the MBCs of NCR169C_17–38_C_12,17_/S-20W^5-Me^ I to a range of 6.3–12.5 µM. Interestingly, the antimicrobial effect of NCR169C_17–38_C_12,17_/S-20W^5-Me^ II was comparable with the parental peptide except for *E. faecalis* (25 µM). These data demonstrated that by substitution of one hydrogen at position 5 of the indole ring with one alkyl group slightly reduced the antibacterial activity. This might be due to the inability of methyl to form a hydrogen bond, as was the case with other molecules containing 5-methyl tryptophan [47,48].

Fluorine with its unique characteristics, such as low polarizability, and high electronegativity, is a promising candidate for peptide development. Currently, more than 20% of drugs on the market are fluorine-containing drugs [49,50,51]. Accordingly, NCR169C_17–38_C_12,17_/S analogs composed of modified fluorotryptophan were the most potent antibacterial agents among all analogs. The killing ability of these peptides demonstrated that analogs containing 6-fluorotryptophan reduced MBCs the most. The MBC values of both NCR169C_17–38_C_12,17_/S-10W^6-F^ I and II range from 0.8 to 1.6 µM, with only one exception (6.3 µM for *S. enterica*). Similarly, peptides with 5-fluorotryptophan, NCR169C_17–38_C_12,17_/S-10W^5-F-L,^ and NCR169C_17–38_C_12,17_/S-10W^5-F-D^, also have MBCs of 1.6—6.3 µM, with the highest MBC for *P. aeruginosa* (Gram-negative) and *L. monocytogenes* (Gram-positive). The increased antibacterial activity of these NCR169C_17–38_C_12,17_/S analogs containing modified fluorotryptophan may be due to the chemical and thermal stability of the peptides, as well as greater resistance of the peptides to proteolysis [52,53,54]. 

The incorporation of one methoxy group to tryptophan did not significantly affect antimicrobial abilities, except that it caused interesting changes in the MBC values of the peptides against *P. aeruginosa*. NCR169C_17–38_C_12,17_/S-10W^5-MeO^ I exhibited an overall antimicrobial activity at 1.6 µM, while NCR169C_17–38_C_12,17_/S-10W^5-MeO^ I and NCR169C_17–38_C_12,17_/S-20W^5-MeO^ at a concentration of 3.1 µM. However, NCR169C_17–38_C_12,17_/S-10W^5-MeO^ I had the highest MBC value of 25 µM against *P. aeruginosa*, while NCR169C_17–38_C_12,17_/S-20W^5-MeO^ possessed an MBC of 0.8 µM, which is the lowest MBC among these analogs for this Gram-negative bacterium. The oxygen in the methoxy group may influence the antibacterial activity by forming a hydrogen bond with the target bacterial membrane. The effect of 5-methoxy tryptophan on peptide activity is reported for instance, on an argyrin analog that contained modified 5-methoxy tryptophan and was active against *P. aeruginosa* and *Proteus mirabilis* [48].

Contrary to expectations, the presence of 7-Aza tryptophan did not enhance the biological activity of the NCR169C_17–38_C_12,17_/S analogs. This finding is interesting since it was presumed that 7-Aza tryptophan, compared to unmodified tryptophan, has more capacity to create hydrogen bonds [55]. However, the presence of 7-Aza tryptophan in peptides or proteins may decrease the activity by the interference of the 7-Aza with the side chain of other amino acids, and the substitution of one carbon at position 7 in the indole ring with nitrogen reduces the hydrophobicity of the amino acid, hence decrease the activity of the peptide [56,57].

Among the presented short and chemically modified peptide derivatives of NCR169, there are several that can kill all or some of the tested bacterial species with a very low MBC (0.8–3.1 µM). In comparison, the MBCs of carbenicillin were 2–4-fold higher, while the MBCs of levofloxacin were either similar or 200-fold higher for the same bacterial species [28]. On the other hand, none of the four active peptides tested had hemolytic activity at the MBC values, and only one of them showed signs of hemolytic activity at higher concentrations, which is promising in terms of the possible later application. At present, the mechanism of action of NCR169 and its derivates needs to be elucidated. An in-depth investigation of the mechanism of active NCR169C derivates from this series would be crucial in understanding how these analogs function in peptide-microbe interactions. The great advantage of these peptides is that they are not cytotoxic to human cells, unlike many other AMPs. This study provides highly potent peptide antimicrobials and enhanced activity by substitution of tryptophan with commercially modified tryptophan at a specific position. 

## 4. Materials and Methods 

### 4.1. Chemical Synthesis of Peptides

All amino acid derivatives were purchased from Chem-Impex (Wood Dale, IL, USA), Iris Biotech GMBH (Marktredwitz, Germany), or Bachem AG (Bubendorf, Switzerland). TentaGel SRAM resin was obtained from Rapp Polymere GmbH (Tübingen, Germany), and anhydrous 1-hydroxybenzotriazole (HOBt) from Abcr GmbH (Karlsruhe, Germany). Solvents and reagents for microwave-assisted peptide synthesis were purchased from the following suppliers: N-methylpyrrolidone (NMP) from Iris Biotech GMBH (Marktredwitz, Germany), dimethylformamide (DMF) from Merck KGaA (Darmstadt, Germany), diisopropylcarbodiimide (DIC) and Oxyma from Fluorochem Ltd. (Hadfield Derbyshire, UK). Solvents and reagents for manual solid-phase synthesis were obtained from the following companies: dichloromethane (DCM), dimethylformamide (DMF), methanol, and piperazine from Alfa Aesar (Thermo Fisher Scientific GmbH, Kandel, Germany), and trifluoroacetic acid (TFA) and dithiothreitol (DTT) from Fluorochem Ltd. (Hadfield Derbyshire, UK). HPLC grade TFA and acetonitrile (AcN) were obtained from Sigma-Aldrich (St. Louis, MO, USA). All other chemicals used were of the highest grade available. All the peptides in Table 1A and Table 3 were synthesized by the standard solid-phase peptide synthesis (SPPS) method on TentaGel S Ram resin (loading 0.23 mmol/g) using an automatic peptide synthesizer (CEM Liberty Blue). The applied chemistry utilized the fluorenyl-9-methoxycarbonyl (Fmoc) amino acid protecting group and diisoproplycarbodimiide/oxyma coupling agent with a 5-fold excess of reagents. Deprotection of the Fmoc group was performed with 10% piperazine and 0.1 mol 1-hydroxy-benzotriazole (HOBt) dissolved in 10% ethanol and 90% dimethylformamide (DMF) in two cycles. The final peptide was cleaved from the resin using a cleavage cocktail consisting of 95:5 (v/V) trifluoroacetic acid (TFA)/water mixture, plus 3% (*w/v*) dithiothreitol (DTT) and 3% (*w/v*) triisopropylsilane (TIS) at room temperature for 3 hr. The resin was removed by filtration and the crude peptides were precipitated by cold diethyl ether. The precipitate was then filtered, and dissolved in water, followed by lyophilization. The crude peptides were analyzed and purified by reverse-phase high-performance liquid chromatography (RP-HPLC), on a 250 × 10 mm C18 column with a solvent system of (A) 0.1% (*v/v*) TFA in water and (B) 80% (*v/v*) acetonitrile and 0.1% TFA (*v/v*) in water at a flow rate of 3.0 mL/min. During this procedure, the different diastereomers could be separated successfully. The purified peptides were characterized by electrospray ionization mass spectroscopy (ESI–MS) using a Waters SQ detector coupled with an Agilent 1200 HPLC system. The column used for this LC–MS analysis was a Luna 5 µ 250 × 4.60 mm C8 column with a solvent system of (A) 0.1% (*v/v*)TFA in water and (B) 80% (*v/v*) acetonitrile and 0.1% TFA in water at a flow rate of 1 mL/min, and the capillary voltage was 3.51 volt. The ESI–MS method allows for multiple charging peptides analysis, hence this technique is a beneficial tool for peptide molecular weight analysis [58,59,60].

### 4.2. Bacterial Strains

The bacterial strains for the antimicrobial assay were obtained from the ATCC (American Type Culture Collection, Manassas, VA, USA) and NCTC (National Collection of Type Cultures, Salisbury, UK). Gram-negative strains were *Pseudomonas aeruginosa* (ATCC 27853), *Escherichia coli* (ATCC 8739, ATCC 35218, and ATCC 25922), *Salmonella enterica* (ATCC 13076), *Klebsiella pneumoniae* (NCTC 13440), *Acinetobacter baumannii* (ATCC 17978) and the Gram-positive strains were *Enterococcus faecalis* (ATCC 29212), *Listeria monocytogenes* (ATCC 19111), and *Staphylococcus aureus* (HNCMO112011 and ATCC 25923). 

### 4.3. Antimicrobial Activity 

Determination of antibacterial activity for the peptides was carried out by utilizing a minimum bactericidal concentration (MBC) bioassay [28]. Synthesized peptides were dissolved in sterile water and used from 25 to 0.125 μM in 2-fold dilution series and incubated with ~10^7^ log phase bacteria in Potassium-Phosphate Buffer (PPB, pH: 7.4) for three hours as described earlier [28]. After that, 5 μl of each sample was placed on LB agar and the growth of bacteria was monitored after overnight incubation at 37 °C. The lowest concentration of the antimicrobial agents that completely eliminated viable bacteria was considered the minimal bactericidal concentration (MBC).

### 4.4. Hemolysis Assay

Human blood was purchased from the Regional Blood Centre in Szeged. The use of human blood for the hemolysis assay has been authorized by the Regional Hungarian Ethics Committee and approved by the Ethics Review Sector of DG RTD (European Commission) in connection with EK’s ERC AdG SymBiotics. The protocol was performed as described by Lima et al. [30]. Baseline OD_560_ values were determined with cells in TBS buffer, while 0.5% Triton X-100 (Serva) was added to the cells at the same time as NCRs represented 100% of hemolysis. The hemolytic activity was calculated as % of red blood cell disruption relative to the positive control sample lysed with detergent Triton X-100.

## Figures and Tables

**Figure 1 ijms-24-02694-f001:**
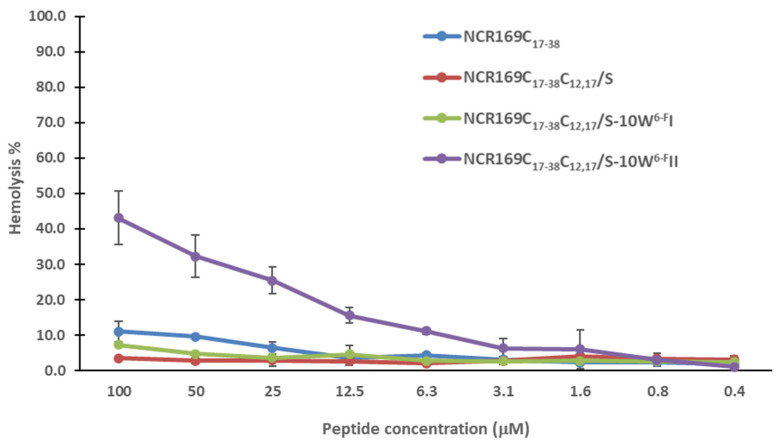
Hemolysis activity of NCR169C_17–38_ and its most active derivatives.

**Table 1 ijms-24-02694-t001:** The antimicrobial activity of NCR169C peptide and its substitution derivatives. A. Amino acid sequence and molecular mass values of the short peptide fragments. Substituted amino acids are underlined. B. Isoelectric point (pI), hydrophobicity (%), grand average hydropathy (GRAVY), and the Boman index (kcal/mol) of the synthesized peptides. C. Minimal bactericidal concentrations (MBC, in μM) of peptides against *E. f.*, *Enterococcus faecalis* (ATCC 29212); *S. a.*, *Staphylococcus aureus* (HNCMO112011); *K. p.*, *Klebsiella pneumoniae* (NCTC 13440); *A. b.*, *Acinetobacter baumannii* (ATCC 17978); *P. a.*, *Pseudomonas aeruginosa* (ATCC 27853); *E. c.*, *Escherichia coli* (ATCC 8739); *L. m.*, *Listeria monocytogenes* (ATCC 19111); *S. e.*, *Salmonella enterica* (ATCC 13076). D. MBCs (μM) of peptides against additional strains of *E. coli* (ATCC 25922 and ATCC 35218) and *S. aureus* (ATCC 25923).

**A.**								
**Peptides**	**Amino Acid Sequence**				**Molecular Mass Calculated**		**Molecular Mass Experimental ^1^**	
NCR169C_17–38_	KSKKPLFKIWKCVENVCVLWYK				2739.4		913.65 (M + 3H)^3+^	
NCR169C_17–38_C_12,17_/S	KSKKPLFKIWKSVENVSVLWYK				2707.3		903.6 (M + 3H)^3+^	
NCR169C_17–38_W_10,20_/A	KSKKPLFKIAKCVENVCVLAYK				2509.1		837.3 (M + 3H)^3+^	
NCR169C_17–38_W_10,20_/A,C_12,17_/S	KSKKPLFKIAKSVENVSVLAYK				2477		826.8 (M + 3H)^3+^	
NCR169C_17–38_W_10,20_C_12,17_/A	KSKKPLFKIAKAVENVAVLAYK				2445		815.7 (M + 3H)^3+^	
**B.**								
**Peptides**	**pI ^2^**		**Hydrophobicity ^2^**		**GRAVY ^2^**		**Boman Index ^3^ (kcal/mol)**	
NCR169C_17–38_	10.1		48.27		−0.15		0.6	
NCR169C_17–38_C_12,17_/S	10.6		47.64		−0.45		1.02	
NCR169C_17–38_W_10,20_/A	10.1		38.74		0.09		0.64	
NCR169C_17–38_W_10,20_/A,C_12,17_/S	10.6		38.04		−0.21		1.07	
NCR169C_17–38_W_10,20_C_12,17_/A	10.6		41.04		0.03		0.59	
**C.**								
**Peptides**	** *E. f.* **	** *S. a.* **	** *K. p.* **	** *A. b.* **	** *P. a.* **	** *E. c.* **	** *L. m.* **	** *S. e.* **
NCR169C_17–38_	6.3	3.1	3.1	3.1	3.1	1.6	3.1	3.1
NCR169C_17–38_C_12,17_/S	3.1	1.6	3.1	1.6	3.1	1.6	3.1	1.6
NCR169C_17–38_W_10,20_/A	-	-	12.5	3.1	3.1	3.1	25	-
NCR169C_17–38_W_10,20_/A,C_12,17_/S	25	6.3	25	12.5	3.1	6.3	3.1	6.3
NCR169C_17–38_W_10,20_C_12,17_/A	-	6.3	25	12.5	3.1	6.3	6.3	6.3
**D.**								
**Peptides**	***E.c.* ATTC 25922**			***E.c.* ATTC 35218**			***S.a.* ATTC 25923**	
NCR169C_17–38_	1.6			1.6			3.1	
NCR169C_17–38_C_12,17_/S	1.6			3.1			3.1	

^1^ (M + 3H)^3+^ refers to the experimentally determined molecular mass of the triple-charged peptide, consistent with the calculated mass; ^2^ values from https://www.thermofisher.com/hu/en/home/life-science/protein-biology/peptides-proteins/custom-peptide-synthesis-services/peptide-analyzing-tool.html (accessed on 20 November 2022); ^3^ values from https://aps.unmc.edu/prediction/predict (accessed on 20 November 2022); -: inactive up to 25 μM.

**Table 2 ijms-24-02694-t002:** List of NCR169C_17–38_C_12,17_/S Analogs.

Peptide Analogs	Amino Acid Sequence	Molecular Mass Calculated	Molecular Mass Experimental
NCR169C_17–38_C_12,17_/S-10W^5-Me^ I *	KSKKPLFKIW^5-Me^KSVENVSVLWYK	2721.26	907.6 (M + 3H)^3+^
NCR169C_17–38_C_12,17_/S-10W^5-Me^ II *	KSKKPLFKIW^5-Me^KSVENVSVLWYK	2721.26	907.8 (M + 3H)^3+^
NCR169C_17–38_C_12,17_/S-20W^5-Me^ I	KSKKPLFKIWKSVENVSVLW^5-Me^ YK	2721.26	908.8 (M + 3H)^3+^
NCR169C_17–38_C_12,17_/S-20W^5-Me^ II	KSKKPLFKIWKSVENVSVLW^5-Me^ YK	2721.26	907.8 (M + 3H)^3+^
NCR169C_17–38_C_12,17_/S-10W^5-F-L^	KSKKPLFKIW^5-F-L^ KSVENVSVLWYK	2725.47	909.6 (M + 3H)^3+^
NCR169C_17–38_C_12,17_/S-10W^5-F-D^	KSKKPLFKIW^5-F-D^ KSVENVSVLWYK	2725.47	909.7 (M + 3H)^3+^
NCR169C_17–38_C_12,17_/S-20W^5-F-L^	KSKKPLFKIWKSVENVSVLW^5-F-L^YK	2725.47	908.9 (M + 3H)^3+^
NCR169C_17–38_C_12,17_/S-10W^6-F^I	KSKKPLFKIW^6-F^KSVENVSVLWYK	2725.47	909.3 (M + 3H)^3+^
NCR169C_17–38_C_12,17_/S-10W^6-F^II	KSKKPLFKIW^6-F^KSVENVSVLWYK	2725.47	909.4 (M + 3H)^3+^
NCR169C_17–38_C_12,17_/S-10W^7-Aza^ I	KSKKPLFKIW^7-Aza^KSVENVSVLWYK	2706.46	903.3 (M + 3H)^3+^
NCR169C_17–38_C_12,17_/S-10W^7-Aza^ II	KSKKPLFKIW^7-Aza^KSVENVSVLWYK	2706.46	903.6 (M + 3H)^3+^
NCR169C_17–38_C_12,17_/S-20W^7-Aza^ I	KSKKPLFKIWKSVENVSVLW^7-Aza^YK	2706.46	903.0 (M + 3H)^3+^
NCR169C_17–38_C_12,17_/S-20W^7-Aza^ II	KSKKPLFKIWKSVENVSVLW^7-Aza^YK	2706.46	903.2 (M + 3H)^3+^
NCR169C_17–38_C_12,17_/S-10W^5-MeO^ I	KSKKPLFKIW^5-MeO^KSVENVSVLWYK	2737.49	913.1 (M + 3H)^3+^
NCR169C_17–38_C_12,17_/S-10W^5-MeO^ II	KSKKPLFKIW^5-MeO^KSVENVSVLWYK	2737.49	913.0 (M + 3H)^3+^
NCR169C_17–38_C_12,17_/S-20W^5-MeO^	KSKKPLFKIWKSVENVSVLW^5-MeO^YK	2737.49	913.2 (M + 3H)^3+^

* I and II refer to either d- or l-configurations of the substituted tryptophan-containing peptides since the modified tryptophans were used in racemic forms for the synthesis and the resulted peptides were separated afterward.

**Table 3 ijms-24-02694-t003:** Minimal bactericidal concentrations (MBC; in μM) of the modified peptides on different pathogens after 3 h of treatment in phosphate buffer (PPB). **A.**
*E*. *f.*, *Enterococcus faecalis* (ATCC 29212); *S. a.*, *Staphylococcus aureus* (HNCMO112011); *K. p.*, *Klebsiella pneumoniae* (NCTC 13440); *A. b.*, *Acinetobacter baumannii* (ATCC 17978); *P. a.*, *Pseudomonas aeruginosa* (ATCC 27853); *E. c.*, *Escherichia coli* (ATCC 8739); *L. m.*, *Listeria monocytogenes* (ATCC 19111); *S. e.*, *Salmonella enterica* (ATCC 13076). The two most active peptides are in bold. **B**. MBCs (μM) of peptides against additional strains of *E. coli* (ATCC 25922 and ATCC 35218) and *S. aureus* (ATCC 25923).

**A.**
**Peptides**	** *E. f.* **	** *S. a.* **	** *K. p.* **	** *A. b.* **	** *P. a.* **	** *E. c.* **	** *L. m.* **	** *S. e.* **
NCR169C_17–38_C_12,17_/S-10W^5-Me^ I	3.1	3.1	6.3	6.3	3.1	3.1	3.1	3.1
NCR169C_17–38_C_12,17_/S-10W^5-Me^ II	3.1	3.1	6.3	3.1	3.1	3.1	3.1	3.1
NCR169C_17–38_C_12,17_/S-20W^5-Me^ I	12.5	6.3	12.5	6.3	6.3	6.3	6.3	6.3
NCR169C_17–38_C_12,17_/S-20W^5-Me^ II	25	3.1	3.1	3.1	3.1	1.6	3.1	3.1
NCR169C_17–38_C_12,17_/S-10W^5-F-L^	1.6	1.6	3.1	1.6	1.6	1.6	1.6	1.6
NCR169C_17–38_C_12,17_/S-10W^5-F-D^	1.6	3.1	3.1	1.6	3.1	1.6	6.3	3.1
NCR169C_17–38_C_12,17_/S-20W^5-F-L^	3.1	1.6	3.1	3.1	6.3	3.1	3.1	3.1
**NCR169C_17–38_C_12,17_/S-10W^6-F^ I**	**1.6**	**0.8**	**1.6**	**1.6**	**0.8**	**0.8**	**1.6**	**1.6**
**NCR169C_17–38_C_12,17_/S-10W^6-F^ II**	**0.8**	**0.8**	**1.6**	**0.8**	**1.6**	**0.8**	**1.6**	**6.3**
NCR169C_17–38_C_12,17_/S-10W^7-Aza^ I	6.3	3.1	6.3	3.1	3.1	3.1	3.1	3.1
NCR169C_17–38_C_12,17_/S-10W^7-Aza^ II	3.1	3.1	6.3	6.3	3.1	3.1	3.1	3.1
NCR169C_17–38_C_12,17_/S-20W^7-Aza^ I	25	6.3	6.3	6.3	6.3	3.1	25	6.3
NCR169C_17–38_C_12,17_/S-20W^7-Aza^ II	12.5	12.5	12.5	12.5	12.5	12.5	12.5	12.5
NCR169C_17–38_C_12,17_/S-10W^5-MeO^ I	1.6	1.6	1.6	1.6	1.6	3.1	1.6	1.6
NCR169C_17–38_C_12,17_/S-10W^5-MeO^ II	3.1	3.1	3.1	3.1	25	3.1	3.1	3.1
NCR169C_17–38_C_12,17_/S-20W^5-MeO^	3.1	3.1	3.1	3.1	0.8	3.1	3.1	3.1
**B.**
**Peptides**	***E.c*. ATTC 25922**	***E.c*. ATTC 35218**	***S.a*. ATTC 25923**
NCR169C_17–38_C_12,17_/S-10W^6-F^ I	0.8	0.8	1.6
NCR169C_17–38_C_12,17_/S-10W^6-F^ II_	0.8	0.8	1.6

Bold letters indicate the two peptides with the best antimicrobial activity together with their lowest MBC values.

## Data Availability

Data is contained within the article or Appendix A.

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
