# Peer review of "Enhanced Antibacterial Activity of Substituted Derivatives of NCR169C Peptide"

_ijms, 2023, doi:10.3390/ijms24032694_

Round 1

Reviewer 1 Report (Previous Reviewer 1)

The manuscript has been improved, and the authors have responded well to the modification.

Author Response

We thank Reviewer 1 for the positive evaluation of our manuscript in the revised version.

Reviewer 2 Report (Previous Reviewer 2)

The authors made the suggested changes and improvements, especially in the discussion section.

I think the manuscript has now met publication standard. However, I noted some minor errors in the manuscript:

They are as follows:

-Line 15: “…cationic NCRs (peptides?) have antimicrobial…” I think the word peptides is missing.

Line 21 and 45: “….and….” is in italics

Line 27: “…synthesis….?”

Line 29: “…activity, and….”

Line 49: “…14-fold…”

Line 55: use synonym of “finding”, “discovery/development”, perhaps?

Line 68: change “is 3257” for “was”

Line 74: “The mechanisms of action of AMPs are diverse”?

Line 74: “Mostly, they…”

Line 86: “arginine histidine”

Line 91: “Rhizobium” should be in italics?

Line 107: pI=8.45, it seems to me like this value comes out of the blue, with no context. Personally I prefer the Molecular weight or net charge at pH7.

Line 141: “…did not harm…” the peptide? Harm should be used when describing the effect on cells etc., not on the properties or capacities of molecules, per say. I suggest using another word….

Line 154: I think you mean “were” instead of “are”, since its all past tense.

Line 224: Technically, the use of the word “demolish” was not wrongfully applied, but I recommend using another word.

Figure 1: “Peptide” instead of “peptid”.  Please use the same shade of black for the graph. I would recommend the use of ticks for the x and y axes.

Observations:

Cite AlphaFold, Jumper et al. or include the link.

For future publications, I suggest looking into the importance of hydrophobic moment, more than net hydrophobicity. I recommend the 3 D hydrophobic moment vector https://www.ibg.kit.edu/HM/?page=helix  developed by Reißer et al. it is very useful when experimental data cannot be simply explained by the predicted 3 structures of peptides.

The support vector machine predictor (http://www.camp3.bicnirrh.res.in/predict/) also works well in certain cases. Although, I think it wont be much help for the  

 substituted derivatives.

 You should also take a look at the membranome server (https://membranome.org/about) , we try to relate activity and specificity with binding energy and other parameters that you might find useful.

Author Response

We thank very much for Reviewer 2 for allocating time to review our revised manuscript, and for finding the remaining errors, thus allowing us to correct them. We are very grateful for the valuable suggestions for further analysis of the peptides and their function in our future publications.

I noted some minor errors in the manuscript:

Line 15: “…cationic NCRs (peptides?) have antimicrobial…” I think the word peptides is missing.            - Corrected

Line 21 and 45: “….and….” is in italics   - Corrected

Line 27: “…synthesis….?”         - Corrected

Line 29: “…activity, and….”       - Corrected

Line 49: “…14-fold…”    - Corrected

Line 55: use synonym of “finding”, “discovery/development”, perhaps?   - Corrected

Line 68: change “is 3257” for “was”        - Corrected

Line 74: “The mechanisms of action of AMPs are diverse”?        - Corrected

Line 74: “Mostly, they…”           - Corrected

Line 86: “arginine histidine”       - Corrected

Line 91: “Rhizobium” should be in italics?          - Corrected

Line 107: pI=8.45, it seems to me like this value comes out of the blue, with no context. Personally, I prefer the Molecular weight or net charge at pH7.

            - Corrected

Line 141: “…did not harm…” the peptide? Harm should be used when describing the effect on cells etc., not on the properties or capacities of molecules, per say. I suggest using another word….

            - Corrected

Line 154: I think you mean “were” instead of “are”, since it’s all past tense.         - Corrected

Line 224: Technically, the use of the word “demolish” was not wrongfully applied, but I recommend using another word.

            - Corrected

Figure 1: “Peptide” instead of “peptid”.  Please use the same shade of black for the graph. I would recommend the use of ticks for the x and y axes.

            - Corrected

Cite AlphaFold, Jumper, et al., or include the link.

            - Reference included (46)

Reviewer 3 Report (New Reviewer)

After reviewing the revised article, I recommend this paper to be published.

Author Response

We thank Reviewer 3 for the positive evaluation of our manuscript in the revised version.

This manuscript is a resubmission of an earlier submission. The following is a list of the peer review reports and author responses from that submission.

Round 1

Reviewer 1 Report

The manuscript is very interesting but some comments should be considered, the comment is found in the attached manuscript as sticky notes.

The manuscript needs linguistic editing.

Reviewer 2 Report

Howan et al. presented a manuscript whereby they enhanced antibacterial activity of a peptide by generating substituted tryptophan derivatives.

The design of peptide variants with enhanced activities and the microbial activity assays are not enough for publication of this work. Authors should carry out experiments in  an effrot to  understand how and why these peptides present enhanced activities.

On the other hand, cysteine was not important for AMP activity of the native molecules, thus, the authors should consider presenting 3D structure predictions using alphafold platform (https://colab.research.google.com/github/sokrypton/ColabFold/blob/main/AlphaFold2.ipynb). It would be helpful to present a comparsion of the peptide the wildtype and the serine variants, and may give a brief discussion of structure function relationship, if the authors see it fit. This would also make the general presentation of the paper more interesting.

Reviewer 3 Report

This study investigated analogs of nodule-specific cysteine-rich (NCR) peptides for their killing ability against ESKAPE pathogens such as Enterococcus faecalis, Staphylococcus aureus, and Escherichia coli. Searching for new antibiotic alternatives is a critically important study. However, this research only studied the antimicrobial activity of different analogs, no mechanism or structure studies were performed, and tryptophan modifications are not new designs. Besides, only in vitro evaluations for these AMPs were analyzed. Overall, the study falls short of the current journal.